# The Association between Statins and Liver Cancer Risk in Patients with Heart Failure: A Nationwide Population-Based Cohort Study

**DOI:** 10.3390/cancers15112959

**Published:** 2023-05-29

**Authors:** Meng-Chuan Lu, Chun-Chao Chen, Meng-Ying Lu, Kuan-Jie Lin, Chun-Chih Chiu, Tsung-Yeh Yang, Yu-Ann Fang, William Jian, Ming-Yao Chen, Min-Huei Hsu, Yu-Hsin Lai, Tsung-Lin Yang, Wen-Rui Hao, Ju-Chi Liu

**Affiliations:** 1Division of Gastroenterology, Department of Internal Medicine, Tri-Service General Hospital, National Defense Medical Center, Taipei 11490, Taiwan; progressinglife@gmail.com; 2Division of Cardiology, Department of Internal Medicine, Shuang Ho Hospital, Taipei Medical University, New Taipei City 23561, Taiwan; b101092035@tmu.edu.tw (C.-C.C.); 17257@s.tmu.edu.tw (C.-C.C.); 15535@s.tmu.edu.tw (T.-Y.Y.); 18516@s.tmu.edu.tw (Y.-A.F.); 3Taipei Heart Institute, Taipei Medical University, Taipei 11031, Taiwan; 21514@s.tmu.edu.tw (K.-J.L.); 151017@h.tmu.edu.tw (T.-L.Y.); 4Division of Cardiology, Department of Internal Medicine, School of Medicine, College of Medicine, Taipei Medical University, Taipei 11031, Taiwan; 5Graduate Institute of Medical Sciences, College of Medicine, Taipei Medical University, Taipei 110, Taiwan; 6Division of Cardiology, Department of Internal Medicine, Taitung MacKay Memorial Hospital, Taitung 95054, Taiwan; davielu.tw@yahoo.com.tw; 7Division of Cardiovascular Surgery, Department of Surgery, Shuang Ho Hospital, Taipei Medical University, New Taipei City 23561, Taiwan; 8Department of Emergency, University Hospitals Cleveland Medical Center, Cleveland, OH 44106, USA; william.jian@gmail.com; 9Division of Gastroenterology and Hepatology, Department of Internal Medicine, School of Medicine, College of Medicine, Taipei Medical University, Taipei 11031, Taiwan; u90223@tmu.edu.tw (M.-Y.C.); auron1018@gmail.com (Y.-H.L.); 10TMU Research Center for Digestive Medicine, Taipei Medical University, Taipei 110, Taiwan; 11Division of Gastroenterology and Hepatology, Department of Internal Medicine, Shuang Ho Hospital, New Taipei City 23561, Taiwan; 12Graduate Institute of Data Science, College of Management, Taipei Medical University, Taipei 11031, Taiwan; 701056@tmu.edu.tw; 13Department of Neurosurgery, Shuang Ho Hospital, Taipei Medical University, New Taipei City 23561, Taiwan; 14Division of Cardiology, Department of Internal Medicine and Cardiovascular Research Center, Taipei Medical University Hospital, Taipei 110, Taiwan

**Keywords:** statins, hydrophilic, lipophilic, liver cancer, heart failure

## Abstract

**Simple Summary:**

Heart failure is a major public health challenge with similar risk factors to those of cancer. HMG-CoA reductase inhibitors, also known as statins, are widely prescribed lipid-lowering agents. Chemoprevention has been reported as a pleiotropic effect of statins. We aimed to evaluate the chemoprotective effect of statins on liver cancer in patients with heart failure and to further identify the differences in effectiveness among statin doses and types. The results demonstrated that statins potentially decreased the risk of liver cancer in patients with heart failure in the entire cohort as well as in sex-, age-, and dose-stratified subgroup analyses as compared with the control group. Moreover, both hydrophilic and lipophilic statins showed significant risk reductions. The findings of the present study demonstrate a potential benefit in terms of liver cancer risk for patients with heart failure using statins.

**Abstract:**

Heart failure (HF) and cancer have similar risk factors. HMG-CoA reductase inhibitors, also known as statins, are chemoprotective agents against carcinogenesis. We aimed to evaluate the chemoprotective effects of statins against liver cancer in patients with HF. This cohort study enrolled patients with HF aged ≥20 years between 1 January 2001 and 31 December 2012 from the National Health Insurance Research Database in Taiwan. Each patient was followed to assess liver cancer risk. A total of 25,853 patients with HF were followed for a 12-year period; 7364 patients used statins and 18,489 did not. The liver cancer risk decreased in statin users versus non-users (adjusted hazard ratio (aHR) = 0.26, 95% confidence interval (CI): 0.20–0.33) in the entire cohort in the multivariate regression analysis. In addition, both lipophilic and hydrophilic statins reduced the liver cancer risk in patients with HF (aHR 0.34, 95% CI: 0.26–0.44 and aHR 0.42, 95% CI: 0.28–0.54, respectively). In the sensitivity analysis, statin users in all dose-stratified subgroups had a reduced liver cancer risk regardless of age, sex, comorbidity, or other concomitant drug use. In conclusion, statins may decrease liver cancer risk in patients with HF.

## 1. Introduction

Heart failure (HF) is a major public health challenge and global epidemic [1,2] caused by aging demographics and the increasing prevalence of comorbidities, such as hypertension, diabetes, coronary artery disease, obesity, and atrial fibrillation [3]. Improvements in HF treatment have further extended life expectancy and increased the prevalence of non-cardiac morbidity in patients with HF. Some epidemiological studies have demonstrated that cancer is the major cause of non-cardiac death in patients with HF [4,5,6]. Predisposing conditions such as neurohormonal activation, systemic inflammation, and oxidative stress have been suggested to contribute to both HF and malignancy [7,8].

HMG-CoA reductase inhibitors (statins) are the most widely prescribed lipid-lowering agents and inhibit the mevalonate pathway. Moreover, studies have demonstrated that statins exert pleiotropic effects, such as anti-inflammatory, antioxidant, and immunomodulatory effects [9,10,11,12]. Various experimental and clinical studies have reported that statins are chemoprotective against carcinogenesis because of their potential pleiotropic mechanisms [13,14,15]. However, although statin use reportedly decreases the risk of many cancers [16], the chemoprotective effects of hydrophilic and lipophilic statins on liver cancer have been inconsistent [17,18]

Primary liver cancer is the fourth leading cause of cancer-related deaths [19], and a nationwide study of statin use in patients with HF reported a lower risk of cancer incidence and cancer-related mortality [20]. In the present study, we aimed to investigate the liver cancer risk in patients with HF with or without statin treatment as well as the differences among statin doses and types.

## 2. Materials and Methods

Taiwan’s National Health Insurance program has been mandatory for all citizens since 1995. The program provides universal health insurance coverage to Taiwanese residents [21]. Data obtained from the National Health Insurance Research Database are similar to those of the general population in regard to age, gender, and health-care costs. Data are anonymized before being released to researchers.

Patients who received a diagnosis of HF (with the International Classification of Diseases, Ninth Revision, Clinical Modification code 428.X; *n* = 45,153) who had at least two outpatient department visits or one hospital admission between 1 January 2001 and 31 December 2012 were identified from the database (*n* = 35,043). Patients who were younger than 40 years (*n* = 1509), who had a history of any cancer before the enrollment date (*n* = 4155) or who had received a statin prescription within 6 months before the enrollment date (*n* = 3526) were excluded. A total of 25,853 patients with HF were included in the study cohort and were followed up for 12 years. Of the 25,853 patients with HF, 7364 used statins and 18,489 did not (Figure 1).

The primary outcome was the incidence of primary liver cancer (with the International Classification of Diseases, Ninth Revision, Clinical Modification code 155.X) during the follow-up period. Follow-up ended on 31 December 2012 or upon receipt of a new diagnosis of liver cancer, withdrawal from the National Health Insurance program, loss to follow-up, or death, whichever occurred first. Demographic characteristics (age and sex), comorbidities (diabetes, hypertension, and dyslipidemia), Charlson comorbidity index (CCI) scores, urbanization level, monthly income, and use of non-statin lipid-lowering drugs (metformin, aspirin, and angiotensin-converting enzyme inhibitors (ACEIs)/angiotensin II receptor blockers (ARBs)) were collected.

We aimed to evaluate the preventive effects of statins in patients with HF who have a higher risk of liver cancer. The end point was primary liver cancer. To measure statin exposure, we used the defined daily dose (DDD) as a measurement tool as defined by the World Health Organization to assume the average maintenance dose per day of a drug consumed for its main indication in adults [22]. A prescription refill lasts for 3 months and can be filled up to three times; therefore, we categorized the DDDs (for the entire observation period for each patient) of statins into four groups in each cohort (<28, 28–90, 91–365, and >365 cumulative DDDs (cDDDs)) to examine the dose–response relationship. Patients with <28 cDDDs who received statins were defined as statin non-users [23]. Furthermore, to compare the effect of the solubility difference of each statin, we categorized statin use into individual statin groups in each cohort to evaluate the preventive effects of different statins.

To estimate the effect of statins, a propensity score (PS) was derived using a logistic regression model accounting for covariates predicting statin exposure. All potential confounders were included in the regressor list. This method has been used in observational studies to reduce selection bias [24]. The covariates in the main model were adjusted for the PSs for age (40–54, 55–64, and ≥75 years), sex, CCI score, diabetes, hypertension, dyslipidemia, urbanization level, and monthly income level (0, 1–21,000, 21,000–33,300, and ≥33,301 New Taiwan dollars) [25]. The Kaplan–Meier method was used for investigating the liver-cancer-free survival rate in patients with HF who were stratified according to statin use status and statin use dose status.

A Cox proportional hazards model was used to calculate the hazard ratios (HRs) for liver cancer among the statin users and non-users. In the multivariate analysis, the HRs were adjusted for age, sex, CCI score, diabetes, hypertension, dyslipidemia, urbanization level, and monthly income. A stratified analysis was conducted to evaluate the effect of statin use on age and sex. All analyses were conducted using SAS software (version 9.3; SAS, Cary, NC, USA); two-tailed *p* < 0.05 was considered significant. In epidemiological studies, through external adjustments, sensitivity analyses can be used to clarify the effects of drugs and other covariates [26].

Therefore, in the present study, sensitivity analyses with adjustments were used to determine the associations of age and sex; diabetes, dyslipidemia, hypertension, and CCI score; and the use of non-statin lipid-lowering drugs (metformin, aspirin, and ACEIs/ARBs) with the incidence of liver cancer. In addition to the covariates in the main model, the models were stratified by the use of different drugs as additional covariates.

## 3. Results

The demographic characteristics, medical conditions, medication use, level of urbanization, and monthly income level of the entire cohort and for patients with HF with and without statin use are listed in Table 1. In total, 25,853 patients with HF were enrolled in the study cohort: 7364 (28.5%) patients with HF used statins and 18,489 (71.5%) did not. The prevalence of pre-existing medical comorbidities, namely hypertension (76.58% versus 72.13%, *p* < 0.001), diabetes (39.42% versus 31.50%, *p* < 0.001), and dyslipidemia (44.91% versus 28.06%, *p* < 0.001) was higher among statin users versus non-users. The percentage of patients with hepatitis B or hepatitis C was lower among statin users than non-users (91.12% versus 88.59%, *p* < 0.001). In addition, several significant differences were observed in the distribution of age, sex, CCI score, monthly income level, urbanization level, and non-statin lipid-lowering drug use, such as metformin, a renin–angiotensin-aldosterone system inhibitor (RAASI), or aspirin, between statin users and non-users.

The distribution of statin use in patients with HF is shown in Table 2. In total, 17,973 (69.52%) patients never used statins, and 516 (2.00%) patients used statins for less than 28 days; 7364 (28.48%) patients used statins for more than 28 days. Among the participants who had ever used statins, 3495 (44.35%) patients used more than one type of statin. The most prescribed statin was atorvastatin (54.11%), followed by rosuvastatin (28.22%) and simvastatin (28.05%).

The liver cancer risk among statin users and non-users in the study cohort is described in Table 3 and Figure 2. The total follow-up durations were 51,080.5 and 85,867.4 person–years for statin users and non-users, respectively. After adjusting for age, sex, CCI score, diabetes, hypertension, dyslipidemia, urbanization level, and monthly income level using PS matching, the risk of liver cancer was analyzed. Compared with patients with HF without statin use, the adjusted HRs (aHRs) for liver cancer risk were decreased in patients with HF who used statins (aHR 0.26, 95% CI [0.20, 0.33]). Further stratified analyses revealed that the aHRs remained significantly decreased in patients with HF who used statins, regardless of age or sex. Compared with the patients with HF without statin use, the aHRs for liver cancer decreased among statin users aged 40–64, 65–74, and ≥75 years (aHR 0.23, 95% CI [0.15,0.34]; aHR 0.30, 95% CI [0.20,0.44]; and 0.25 [0.13, 0.48], respectively). In the sex-stratified analysis, the aHRs for liver cancer were lower in patients with HF versus those without statin use (women: aHR 0.25, 95% CI [0.17, 0.36]; men: aHR 0.27, 95% CI [0.19, 0.38]).

The incidence rates and aHRs for liver cancer associated with different statin doses during the follow-up period are listed in Table 4 and Figure 3. Compared with the non-users, statin users had a lower risk of liver cancer (aHR 0.26, 95% CI [0.20, 0.33]). Subgroup analysis of different statin doses, namely 28–90, 91–365, and >365 cDDDs, were associated with a lower liver cancer risk compared with non-users (aHR 0.42, 95% CI [0.27, 0.67]; aHR 0.32,95% CI [0.21, 0.48]; and aHR 0.18, 95% CI [0.12, 0.26], respectively). Statins can be classified as lipophilic or hydrophilic based on their solubility in water. Lipophilic statins include simvastatin, lovastatin, atorvastatin, and fluvastatin; hydrophilic statins include pravastatin and rosuvastatin. Both lipophilic and hydrophilic statins reduced the liver cancer risk in patients with HF (aHR 0.34, 95% CI [0.26,0.44] and aHR 0.42, 95% CI [0.28,0.64], respectively). All subgroups at each statin dose showed a decreased risk of liver cancer, except for the group with 28–90 cDDDs among hydrophilic statin users. Subgroup analyses of individual statins revealed a significantly lower risk of liver cancer in patients treated with simvastatin, atorvastatin, and rosuvastatin alone (aHRs: 0.51, 0.39, and 0.34, respectively).

Table 5 presents the sensitivity analyses results examining the association between statin treatment and reduced liver cancer risk as measured by aHRs. To estimate the effect of statin dose on liver cancer risk reduction, the entire cohort was treated with <28, 28–90, 91–365, or >365 cDDDs. Additional covariate adjustments were made to the main model, including non-statin medication, metformin, RAASI, and aspirin use, to assess the association of statin use with the risk of liver cancer for the four potential chemoprotective drugs separately. The results showed a significantly lower risk of liver cancer compared with the nonusers in the main model with the additional covariates, with a tendency towards dose-dependent risk reduction. In addition, subgroup analyses by age, sex, CCI score, diabetes, dyslipidemia, hypertension, hepatitis B or C, non-statin lipid-lowering drugs, metformin, RAASI, and aspirin were conducted. All aHRs indicated that the statins in each dose-stratified group significantly reduced the liver cancer risk in all subgroups, regardless of age, sex, comorbidities, or concomitant drug use, as compared with that in the statin group with <28 cDDDs.

## 4. Discussion

To the best of our knowledge, this is the first study to assess the effect of statins on the reduction of liver cancer risk in patients with HF. The effects of different statin types and doses were also investigated. Our results demonstrated that statin users had a significantly lower risk of liver cancer than nonusers among patients with HF in the entire cohort and in the sex- and age-stratified subgroup analyses. A significant risk reduction was also noted in the dose-stratified subgroups of 28–90, 91–365, and > 365 cDDDs. A potential dose-dependent effect of statins on liver disease risk reduction was observed. In addition, both hydrophilic and lipophilic statin use significantly decreased the liver cancer risk compared with that in non-users among patients with HF.

HF and cancer share common pathophysiological mechanisms [27,28,29], and several studies have indicated that patients with HF are prone to carcinogenesis [7,30,31,32] derived from a number of predisposing conditions of malignancy, including neurohormonal activation to tumorigenesis, systemic inflammation, and oxidative stress [7,8]. Meanwhile, HMG-CoA reductase inhibitors (statins) and lipid-lowering agents [33] also exert anti-inflammatory [34], antioxidative [35], and anti-cancer effects [14,36]. Statin use may confer chemoprotective effects through various mechanisms, including the inhibition of downstream products in the mevalonate pathway [9,10,11,37,38], triggering tumor-specific apoptosis [39], arresting the growth 1 phase of the cell cycle by inhibiting the proteasome pathway [40], reversing the likelihood of malignancy, and reducing the invasiveness of carcinoma in situ [38]. The dominant anti-inflammatory effects of statins in patients with HF were reported in a previous study [34], which implied the potential of diminishing cancer progression in this patient group. A retrospective cohort study revealed that statins were associated with a lower liver cancer risk in patients with HF in a subgroup analysis [20]. Our study is the first to demonstrate that statin use is associated with a decreased risk of liver cancer in patients with HF compared with patients without statin use based on the analysis of a nationwide database, independent of sex and age.

Our study assessed the effect of statin dose on liver cancer risk reduction and revealed that statin use had a tendency towards dose-dependent risk reduction among dose-stratified groups in the entire cohort and by statin cohorts. Nevertheless, in the subgroup analysis of metformin, among the subgroup of metformin use of less than 28 cDDDs, all statin use > 28 cDDDs decreased the liver cancer risk, whereas statin use of more than 365 cDDDs was associated with a significantly lower risk of liver cancer than that of statin use of 28–90 cDDDs, revealing a dose-dependent effect (aHR 0.50, CI [0.30–0.84], aHR 0.15, [0.08–0.27] for 28–90 and >365 cDDDs, respectively). Meanwhile, among the subgroup of metformin use of > 365 cDDDs, statin use of 28–90 cDDDs did not significantly reduce the risk of liver cancer (aHR 0.27, CI [0.07–1.11]). However, after a longer period of statin usage, a significantly lower risk was observed among patients with metformin use > 365 cDDDs (aHR 0.27, CI [0.11–0.67] and aHR 0.22, CI [0.12–0.41] for statin use 91–365 cDDDs and >365 cDDDs respectively). Metformin is one of the standard treatments for patients with diabetes. It has been reported before that diabetes is associated with an increased risk of cancer occurrence [41]. Therefore, patients with long-duration metformin use may need longer statin use to exhibit potential chemoprotective effects. Similarly, a case–control study from a nationwide population-based database evaluated the effect of 15 exposure combinations comprising four common drugs (statins, aspirin, metformin, and ACEIs/ARBs) on chemoprotective effects in comparison with a non-exposure group. The results demonstrated that individual or concomitant use of statins, aspirin, and ACEIs/ARBs could reduce the risk of liver cancer compared with the non-exposure group, although metformin use or concomitant use with metformin may increase the risk of liver cancer [42].

Consensus on the chemoprotective effects of hydrophilic and lipophilic statins in reducing the risk of liver cancer remains controversial [17,18]. Lipophilic statins enter cells predominantly via passive diffusion and are widely distributed in tissues, whereas the uptake of hydrophilic statins involves a liver-specific carrier-mediated mechanism [43]. For hepatitis virus carriers, interruption of the mevalonate pathway prevents viral replication by potentiating antiviral therapy and stimulating anti-tumor immunity [44,45,46,47,48]. Most nationwide studies and meta-analyses have shown that lipophilic statins reduce the risk of liver cancer; however, the results for hydrophilic statins have been inconsistent [15,18,49]. In our study, two hydrophilic statins were evaluated, with rosuvastatin being the most commonly used hydrophilic statin in our cohort (28.22%), followed by pravastatin (12.91%). Our findings showed that hydrophilic statins reduced the risk of liver cancer; however, in our single-drug analysis, only rosuvastatin significantly reduced the risk of liver cancer. Similar results were reported in two other meta-analysis studies [18,50]. Rosuvastatin has a more potent affinity for the active site of HMG-CoA reductase than other statins, and the hepatic uptake of rosuvastatin is reportedly more selective and efficient compared with other drugs [51,52,53,54]. Hence, these features potentially make rosuvastatin different from other hydrophilic statins and exhibit a strong chemoprotective effect in reducing the risk of liver cancer. However, as a high-potency statin [55], rosuvastatin may be more commonly indicated for patients with higher cholesterol levels and more severe heart failure, introducing bias by indication. Future studies are warranted to validate the findings of the present study.

Despite efforts to balance confounding factors, the present study has several limitations. First, this study was conducted using data from a health insurance claims database that lacks information on certain liver cancer risk factors, such as alcohol consumption, smoking habits, aflatoxin exposure, body mass index, and atherosclerosis [56,57]. Hence, we were unable to control for these potential confounding factors. However, we used PS matching to match patients by age, sex, CCI score, diabetes, hypertension, dyslipidemia, urbanization level, and monthly income. Urbanization level and monthly income were alternative factors for lifestyle and environmental factors [25,58]. Moreover, CCI scores include myocardial infarction, peripheral vascular disease, and cerebrovascular disease, which have a strong relationship with atherosclerosis. Second, we had no access to laboratory data or details of medical treatments for patients with hyperlipidemia. However, we conducted a sensitivity analysis of the adjusted HRs by dose-exposure-stratified subgroup analysis, which revealed persistent effectiveness trends in the main mode with additional covariates. Further investigations of the relationship between the effectiveness and dosage of statins may be considered to assist clinical novelty in real-world practice. Third, this was not a prospective, randomized control study. Although our study demonstrated significant results, further research is warranted to investigate the precise cause–effect relationship between statin use and liver cancer in patients with HF. Fourth, the HF patients were enrolled through the codes of International Classification of Diseases, Ninth Revision, which could not reveal the etiology of heart failure, such as cardiogenic or non-cardiogenic HF. Finally, the main factors underlying liver cancer in Taiwan differ from those in Western countries. Thus, the transferability of our findings to other healthcare systems may not be feasible.

## 5. Conclusions

This is the first nationwide population-based cohort study to investigate the effects of statins on liver cancer risk in patients with HF. Our study revealed that statin use is associated with decreased liver cancer risk in patients with HF in the entire cohort, as well as in sex- and age-stratified subgroup analyses, compared with participants without statin use. A reduction in liver cancer risk was also observed in each dose-stratified subgroup. In addition, both hydrophilic and lipophilic statins showed significant risk reduction.

## Figures and Tables

**Figure 1 cancers-15-02959-f001:**
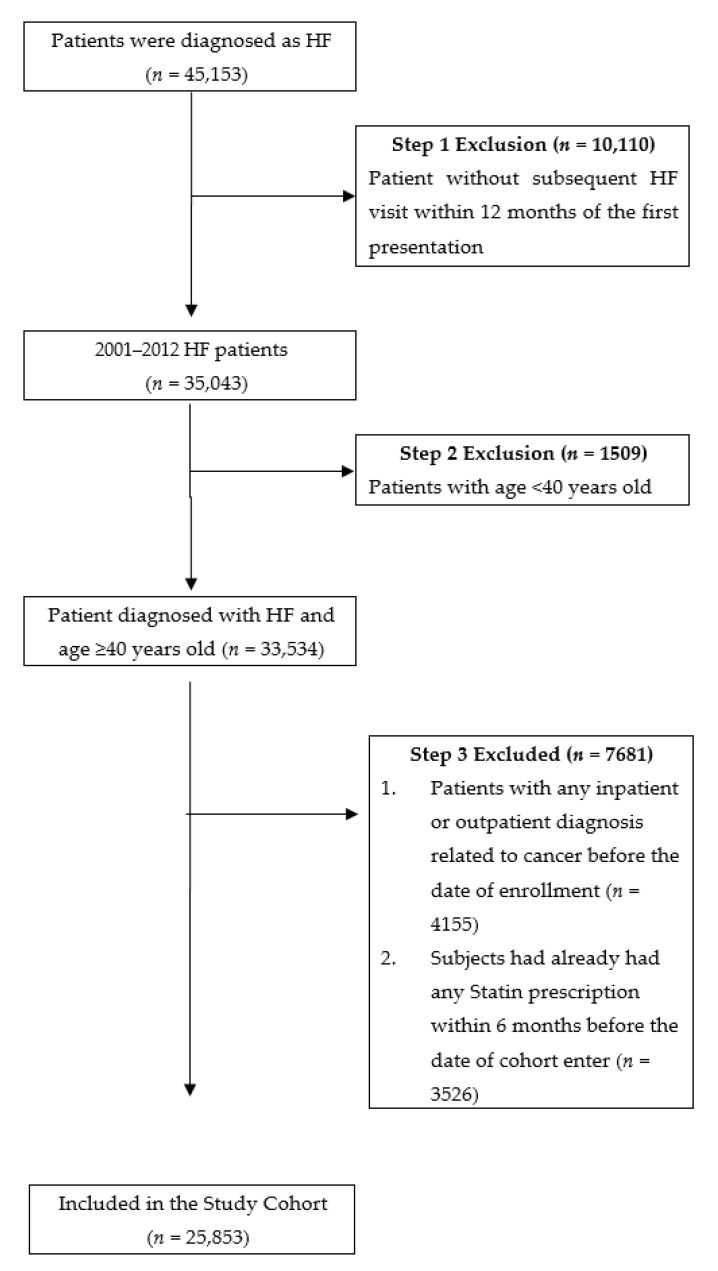
Data selection flowchart. During 2001 to 2012, there were 45,153 patients with HF diagnoses enrolled in this study. Patients diagnosed with HF without subsequent outpatient or inpatient visits, aged less than 40 years, or with pre-existing cancer or a history of statins prescription within 6 months before the date of enrollment were excluded. A total of 25,853 patients were enrolled in the study cohort.

**Figure 2 cancers-15-02959-f002:**
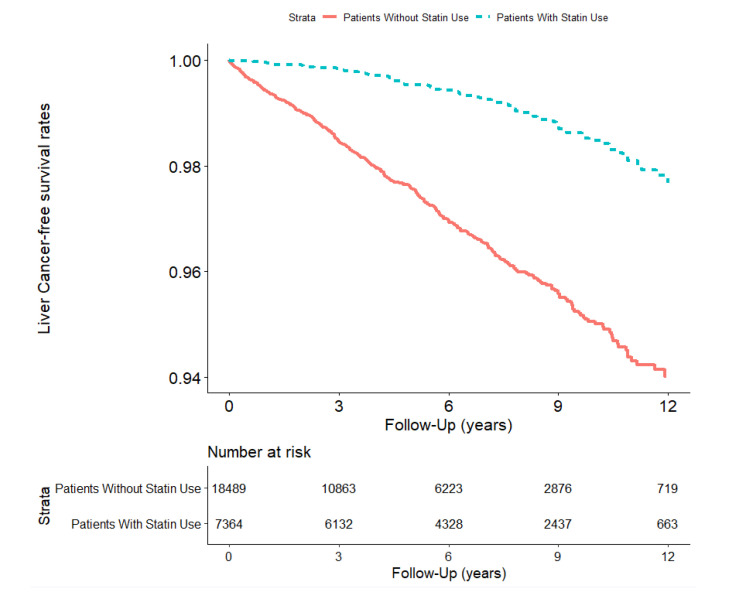
Free of liver cancer survival rate in HF patients in Taiwan (*n* = 25,853) from 1 January 2001 to 31 December 2012 stratified according to statin use status (log–rank test, χ^2^ = 123.135; df = 1; *p* < 0.001).

**Figure 3 cancers-15-02959-f003:**
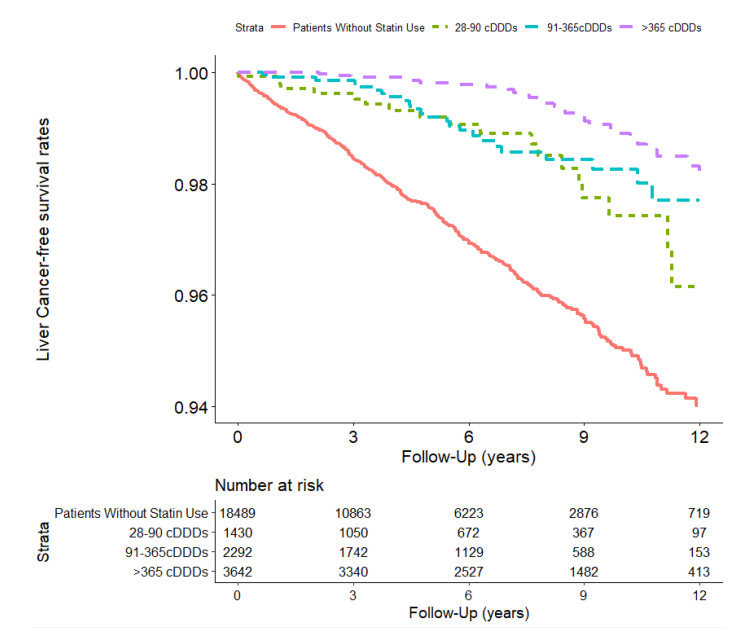
Free of liver cancer survival rate in HF patients in Taiwan (*n* = 25,853) from 1 January 2001 to 31 December 2012 stratified according to statin use dose status (log–rank test, χ^2^ = 126.632; df = 3; *p* < 0.001).

**Table 1 cancers-15-02959-t001:** Characteristic of the sample population.

	Whole Cohort (*n* = 25,853)		Patients with Statin Use (≥28 Days; *n* = 7364)		Patients without Statin Use (<28 Days; *n* = 18,489)		*p* ^a^
	*n*	%	*n*	%	*n*	%	
Age, years (Mean ± SD)	69.71 (13.18)		65.86 (11.15)		71.25 (12.58)		<0.001
40–54	3892	15.05	1428	19.39	2464	13.33	<0.001
55–64	4545	17.58	1776	24.12	2769	14.98	
65–74	7665	29.65	2505	34.02	5160	27.91	
≥75	9751	37.72	1655	22.47	8096	43.79	
Gender							
Female	13,409	51.87	4126	56.03	9283	50.21	<0.001
Male	12,444	48.13	3238	43.97	9206	49.79	
CCI ^+^							
0	5562	21.51	1825	24.78	3737	20.21	<0.001
1	6499	25.14	1997	27.12	4502	24.35	
2	5411	20.93	1523	20.68	3888	21.03	
≥3	8381	32.42	2019	27.42	6362	34.41	
Diabetes							
No	17,126	66.24	4461	60.58	12,665	68.50	<0.001
Yes	8727	33.76	2903	39.42	5824	31.50	
Hypertension							
No	6878	26.60	1725	23.42	5153	27.87	<0.001
Yes	18,975	73.40	5639	76.58	13,336	72.13	
Dyslipidemia							
No	17,358	67.14	4057	55.09	13,301	71.94	<0.001
Yes	8495	32.86	3307	44.91	5188	28.06	
Non-statin lipid-lowering drugs							
<28 days	22,909	88.61	5427	73.70	17,482	94.55	<0.001
28–365 days	2027	7.84	1274	17.30	753	4.07	
>365 days	917	3.55	663	9.00	254	1.37	
Metformin							
<28 days	20,061	77.60	4502	61.14	15,559	84.15	<0.001
28–365 days	2203	8.52	823	11.18	1380	7.46	
>365 days	3589	13.88	2039	27.69	1550	8.38	
RAA							
<28 days	7257	28.07	957	13.00	6300	34.07	<0.001
28–365 days	6906	26.71	1521	20.65	5385	29.13	
>365 days	11,690	45.22	4886	66.35	6804	36.80	
Aspirin							
<28 days	10,706	41.41	1701	23.10	9005	48.70	<0.001
28–365 days	6737	26.06	1873	25.43	4864	26.31	
>365 days	8410	32.53	3790	51.47	4620	24.99	
Level of Urbanization							
Urban	16,521	63.90	4943	67.12	11,578	62.62	<0.001
Suburban	5935	22.96	1542	20.94	4393	23.76	
Rural	3397	13.14	879	11.94	2518	13.62	
Monthly income (TWD)							
0	3275	12.67	791	10.74	2484	13.44	<0.001
1–21,000	6379	24.67	1567	21.28	4812	26.03	
21,000–33,300	10,337	39.98	2888	39.22	7449	40.29	
≥33,301	5862	22.67	2118	28.76	3744	20.25	
Hepatitis B/C							
No	23,090	89.31	6710	91.12	16,380	88.59	<0.001
Yes	2763	10.69	654	8.88	2109	11.41	

^a^ Comparison between non-statin and statin users; CCI ^+^: Charlson comorbidity index.

**Table 2 cancers-15-02959-t002:** Distribution of statin use.

	Whole Cohort (*n* = 25,853)	
	*n*	%
Total cohort		
Never used	17,973	69.52
≤28 days of statin use	516	2.00
>28 days of statin use	7364	28.48
Ever statin users		
Single statin users	4385	55.65
Sequential multiple statins users	3495	44.35
Statin type ever prescribed ^a^		
Simvastatin	2210	28.05
Lovastatin	1200	15.23
Atorvastatin	4264	54.11
Fluvastatin	1197	15.19
Pravastatin	1017	12.91
Rosuvastatin	2224	28.22

^a^ The sum is more than 100% due to sequential multiple statin prescriptions in 44.35% of the ever-used subjects.

**Table 3 cancers-15-02959-t003:** Risk of liver cancer among statin and non-statin users in the study cohort.

All Group (*n* = 25,853)	Patients without Statin Use (Total Follow-Up 85,867.4 Person–Years)			Patients with Statin Use (Total Follow-Up 51,080.5 Person–Years)			Adjusted HR ^†^ (95% CI)
	No. of Patients with Liver Cancer	Incidence Rate (per 10^5^ Person–Years) (95% CI)		No. of Patients with Liver Cancer	Incidence Rate (per 10^5^ Person–Years) (95% CI)		
Whole cohort							
Liver Cancer	441	513.6	(465.6, 561.5)	71	139.0	(106.7, 171.3)	0.26(0.20, 0.33) ***
Age, 40–64 ^a^							
Liver Cancer	153	537.0	(451.9, 622.1)	30	126.9	(81.5, 172.3)	0.23(0.15, 0.34) ***
Age, 65–74 ^b^							
Liver Cancer	160	590.7	(499.2, 682.2)	31	171.6	(111.2, 232.0)	0.30(0.20, 0.44) ***
Age, ≥75 ^c^							
Liver Cancer	128	422.6	(349.4, 495.8)	10	106.6	(40.5, 172.7)	0.25(0.13, 0.48) ***
Female ^d^							
Liver Cancer	202	447.2	(385.5, 508.8)	33	110.0	(72.5, 147.5)	0.25(0.17, 0.36) ***
Male ^e^							
Liver Cancer	239	587.3	(512.9, 661.8)	38	180.3	(123.0, 237.6)	0.27(0.19, 0.38) ***

***: *p* < 0.001.^a^ Total follow-up 28,492.4 and 23,637.0 person–year for patients without statin use and patients with statin use. ^b^ Total follow-up 27,086.6 and 18,062.3 person–year for patients without statin use and patients with statin use. ^c^ Total follow-up 30,288.4 and 9381.2 person–year for patients without statin use and patients with statin use. ^d^ Total follow-up 45,173.6 and 30,001.8 person–year for patients without statin use and patients with statin use. ^e^ Total follow-up 40,693.7 and 21,078.7 person–year for patients without statin use and patients with statin use. C.I.: confidence interval. HR: hazard ratio. ^†^ The main model is adjusted for age, sex, Charlson comorbidity index score, diabetes, hypertension, dyslipidemia, level of urbanization, and monthly income in propensity score.

**Table 4 cancers-15-02959-t004:** Incidence rate and adjusted HRs of liver cancer associated with statin use during the follow-up period in HF patients.

Variable	No. of Patients	No. of Person-Years	No. of Patients with Liver Cancer	Incidence Rate (per 10^5^ Person–Years) (95% CI)		Adjusted HR (95% CI)	*p*-Value for Trend
Total statin use							
Non-user (<28 cDDDs)	18,489	85,867.4	441	513.6	(465.6, 561.5)	1.00	<0.001
User (≥28 cDDDs)	7364	51,080.5	71	139.0	(106.7, 171.3)	0.26(0.20, 0.33) ***	
28–90 cDDDs	1430	8530.1	19	222.7	(122.6, 322.9)	0.42(0.27, 0.67) ***	
91–365 cDDDs	2292	14,038.9	24	171.0	(102.6, 239.3)	0.32(0.21, 0.48) ***	
>365 cDDDs	3642	28,511.5	28	98.2	(61.8, 134.6)	0.18(0.12, 0.26) ***	
Lipophilia statin use ^†^							
Non-user (<28 cDDDs)	19,558	91,882.1	448	487.6	(442.4, 532.7)	1.00	<0.001
User (≥28 cDDDs)	6295	45,065.8	64	142.0	(107.2, 176.8)	0.34(0.26, 0.44) ***	
28–90 cDDDs	1376	8485.9	19	223.9	(123.2, 324.6)	0.49(0.31, 0.78) **	
91–365 cDDDs	2150	14,083.9	20	142.0	(79.8, 204.2)	0.33(0.21, 0.52) ***	
>365 cDDDs	2769	22,495.9	25	111.1	(67.6, 154.7)	0.27(0.18, 0.41) ***	
Hydrophilia statin use ^†^							
Non-user (<28 cDDDs)	18,489	85,867.4	488	568.3	(517.9, 618.7)	1.00	<0.001
User (≥28 cDDDs)	2986	21,378.3	24	112.3	(67.3, 157.2)	0.42(0.28, 0.64) ***	
28–90 cDDDs	779	5168.5	7	135.4	(35.1, 235.8)	0.47(0.22, 1.00)	
91–365 cDDDs	1086	7493.3	9	120.1	(41.6, 198.6)	0.46(0.24, 0.90) *	
>365 cDDDs	1121	8716.5	8	91.8	(28.2, 155.4)	0.35(0.17, 0.71) **	
Individual statin use (≥28 cDDDs) ^‡^							
Simvastatin	2210	17,280.4	22	127.3	(74.1, 180.5)	0.51(0.33, 0.79) **	
Lovastatin	1200	10,016.9	18	179.7	(96.7, 262.7)	0.70(0.43, 1.13)	
Atorvastatin	4264	30,556.4	39	127.6	(87.6, 167.7)	0.39(0.28, 0.55) ***	
Fluvastatin	1197	9276.5	13	140.1	(64.0, 216.3)	0.63(0.36, 1.10)	
Pravastatin	1017	7827.8	13	166.1	(75.8, 256.4)	0.73(0.41, 1.27)	
Rosuvastatin	2224	15,714.8	14	89.1	(42.4, 135.8)	0.34(0.20, 0.58) ***	

* *p* < 0.05, ** *p* < 0.01, *** *p* < 0.001.The covariates in the main model were adjusted for the propensity scores for age, sex, Charlson comorbidity index score, diabetes, hypertension, dyslipidemia, level of urbanization, and monthly income. ^†^ Lipophilia statins include simvastatin, lovastatin, atorvastatin, and fluvastatin. Hydrophilia statins include pravastatin and rosuvastatin. ^‡^ The HRs of individual statin users (≥28 cDDDs) were compared with the total statin use of non-users (<28 cDDDs).

**Table 5 cancers-15-02959-t005:** Sensitivity analysis of adjusted HRs of statins in the reduction of liver cancer risk.

	Statin Use				*p* for Trend
	<28 cDDDs	28–90 cDDDs	91–365 cDDDs	>365 cDDDs	
	Adjusted HR (95% CI)	Adjusted HR (95% CI)	Adjusted HR (95% CI)	Adjusted HR (95% CI)	
Main model ^†^	1.00	0.42 (0.27, 0.67) ***	0.32 (0.21, 0.48) ***	0.18 (0.12, 0.26) ***	<0.001
Additional covariates ^‡^					
Main model + Non-statin	1.00	0.43 (0.27, 0.69) ***	0.34 (0.22, 0.51) ***	0.19 (0.13, 0.29) ***	<0.001
Main model + Metformin	1.00	0.41 (0.26, 0.65) ***	0.31 (0.20, 0.47) ***	0.18 (0.12, 0.27) ***	<0.001
Main model + RAA	1.00	0.44 (0.28, 0.69) ***	0.37 (0.24, 0.56) ***	0.23 (0.15, 0.34) ***	<0.001
Main model + Aspirin	1.00	0.45 (0.29, 0.72) ***	0.37 (0.24, 0.56) ***	0.22 (0.15, 0.33) ***	<0.001
Subgroup effects					
Age, years					
40–64	1.00	0.21 (0.08, 0.57) **	0.23 (0.11, 0.47) ***	0.23 (0.14, 0.38) ***	<0.001
65–74	1.00	0.49 (0.25, 0.97) **	0.46 (0.26, 0.81) ***	0.15 (0.08, 0.30) ***	<0.001
≥75	1.00	0.72 (0.32, 1.63)	0.24 (0.08, 0.75) *	0.05 (0.01, 0.37) **	<0.001
Sex					
Female	1.00	0.48 (0.26, 0.89) *	0.24 (0.12, 0.48) ***	0.17 (0.10, 0.30) ***	<0.001
Male	1.00	0.36 (0.18, 0.73) **	0.39 (0.23, 0.66) ***	0.18 (0.11, 0.31) ***	<0.001
CCI +					
0	1.00	0.30 (0.07, 1.23)	0.41 (0.16, 1.02)	0.32 (0.15, 0.65) **	<0.001
1	1.00	0.14 (0.04, 0.58) **	0.22 (0.09, 0.54) ***	0.13 (0.06, 0.29) ***	<0.001
2	1.00	0.46 (0.19, 1.13)	0.35 (0.16, 0.76) **	0.13 (0.05, 0.32) ***	<0.001
≥3	1.00	0.63 (0.33, 1.21)	0.30 (0.14, 0.63) **	0.17 (0.08, 0.36) ***	<0.001
Diabetes					
No	1.00	0.53 (0.31, 0.90) *	0.37 (0.22, 0.63) ***	0.17 (0.10, 0.30) ***	<0.001
Yes	1.00	0.25 (0.10, 0.62) **	0.23 (0.12, 0.46) ***	0.16 (0.09, 0.29) ***	<0.001
Dyslipidemia					
No	1.00	0.35 (0.19, 0.66) **	0.28 (0.16, 0.50) ***	0.18 (0.11, 0.30) ***	<0.001
Yes	1.00	0.54 (0.27, 1.06)	0.36 (0.20, 0.66) ***	0.17 (0.10, 0.31) ***	<0.001
Hypertension					
No	1.00	0.42 (0.17, 1.04)	0.39 (0.17, 0.89) *	0.31 (0.16, 0.62) ***	<0.001
Yes	1.00	0.42 (0.25, 0.72) **	0.30 (0.18, 0.48) ***	0.14 (0.09, 0.23) ***	<0.001
Hepatitis B/C					
No	1.00	0.55 (0.29, 1.04)	0.42 (0.24, 0.74) **	0.20 (0.11, 0.35) ***	<0.001
Yes	1.00	0.38 (0.19, 0.73) **	0.29 (0.16, 0.53) ***	0.28 (0.17, 0.48) ***	<0.001
Non-Statin					
<28 days	1.00	0.39 (0.23, 0.66) ***	0.33 (0.21, 0.53) ***	0.17 (0.11, 0.27) ***	<0.001
28–365 days	1.00	0.86 (0.32, 2.33)	0.46 (0.17, 1.23)	0.28 (0.12, 0.66) **	0.002
>365 days	1.00		0.22 (0.03, 1.97)	0.29 (0.07, 1.33)	0.123
Metformin					
<28 days	1.00	0.50 (0.30, 0.84) **	0.37 (0.23, 0.62) ***	0.15 (0.08, 0.27) ***	<0.001
28–365 days	1.00	0.22 (0.05, 0.90) *	0.19 (0.06, 0.63) **	0.16 (0.05, 0.51) **	<0.001
>365 days	1.00	0.27 (0.07, 1.11)	0.27 (0.11, 0.67) **	0.22 (0.12, 0.41) ***	<0.001
RAA					
<28 days	1.00	0.68 (0.33, 1.39)	0.53 (0.25, 1.13)	0.17 (0.06, 0.55) **	<0.001
28–365 days	1.00	0.27 (0.10, 0.73) **	0.26 (0.10, 0.63) **	0.27 (0.12, 0.61) **	<0.001
>365 days	1.00	0.45 (0.21, 0.97) *	0.38 (0.21, 0.69) **	0.23 (0.14, 0.38) ***	<0.001
Aspirin					
<28 days	1.00	0.47 (0.22, 0.99) *	0.32 (0.14, 0.71) **	0.20 (0.09, 0.46) ***	<0.001
28–365 days	1.00	0.52 (0.25, 1.06)	0.42 (0.21, 0.84) *	0.24 (0.11, 0.53) ***	<0.001
>365 days	1.00	0.36 (0.13, 0.99) *	0.38 (0.19, 0.76) **	0.23 (0.13, 0.40) ***	<0.001

*: *p* < 0.05, **: *p* < 0.01, ***: *p* < 0.001. HR: hazard ratio. ^+^ CCI: Charlson Comorbidity Index. ^†^ The main model is adjusted for age, sex, Charlson comorbidity index, diabetes, hypertension, dyslipidemia, level of urbanization, and monthly income in propensity score. ^‡^ The models were adjusted for covariates in the main model as well as each additional listed covariate.

## Data Availability

The data supporting the findings of the present research were sourced from the NHIRD in Taiwan. Due to legal restrictions imposed by the government of Taiwan related to the Personal Information Protection Act, the database cannot be made publicly available. However, upon reasonable request to the authors and with permission from the NHIRD, the relevant data are available.

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
