# Peer review of "The Association between Statins and Liver Cancer Risk in Patients with Heart Failure: A Nationwide Population-Based Cohort Study"

_cancers, 2023, doi:10.3390/cancers15112959_

Round 1
Reviewer 1 Report
This study evaluated the chemoprotective effect of statins on liver cancer in patients with heart failure and identify the differences in effectiveness among statin doses and types. The results demonstrated that statins potentially decreased the risk of liver cancer in patients with heart failure in the entire cohort as well as in sex-, age-, and dose-stratified subgroup analyses as compared with the control group.
Overall the presentation is clear and comprehensive. A few concerns need further explanations.
1. Heart failure is a clinical symptom, not a medical diagnosis. The causes of heart failure are quite heterogeneous. How do authors define the underlying diagnosis (cardiogenic vs non-cardiogenic HF) related to treatment?
2. Table 3 is confusing because "All cancer" is studied in this table. Is it a previous template for paper writing?
3. Survival curves showing survival differences between statin users vs nonusers should be provided to be more readable.
4. It is well known that hepatitis B and C are the most important causes of liver cancer worldwide. This information should be provided in the paper. Is there any difference in the prevalence rate of hepatitis B/C between statin users and nonusers?
MInor English editing is suggested to improve the fluency of the paper.
Author Response
- Heart failure is a clinical symptom, not a medical diagnosis. The causes of heart failure are quite heterogeneous. How do authors define the underlying diagnosis (cardiogenic vs non-cardiogenic HF) related to treatment?
Response: Our data were obtained from the National Health Insurance Research Database, and HF patients were enrolled through ICD-9 code 428.X (428.0 congestive heart failure, 428.1 left heart failure, 428.2 systolic heart failure, 428.3 diastolic heart failure, 428.4 combined systolic and diastolic heart failure, and 428.9 heart failure, unspecified). It is our limitation to discuss the underlying diagnosis of heart failure, especially for patients with cardiogenic and non-cardiogenic heart failure. We have listed our manuscript’s limitations (lines 369-372). There is a study discussing the relationship between statin and cancer in HF patients, which also enrolled the HF through the ICD-9 code. (https://pubmed.ncbi.nlm.nih.gov/34157723/) - Table 3 is confusing because "All cancer" is studied in this table. Is it a previous template for paper writing?
Response: Thanks a lot for your kind reminder. We have revised it as suggested.
- Survival curves showing survival differences between statin users vs nonusers should be provided to be more readable.
Response: Thanks a lot for your advice. We have added the survival curve between statin users and statin non-users is presented in Figure 2. Besides,
the survival curve between different statin dosage users is presented in Figure 3. The methods were added in lines 154-156.
- It is well known that hepatitis B and C are the most important causes of liver cancer worldwide. This information should be provided in the paper. Is there any difference in the prevalence rate of hepatitis B/C between statin users and nonusers?
Response: Thanks a lot for your advice. We have added the percentage of HBV/HCV in patients with or without statin use in Table 1, and there is a difference between the two groups. However, compared to patients with statin use < 28 cDDD, patients with statin use more than 90 cDDD have significantly lower liver cancer risk in HBV/HCV or non-HBV/HCV group. The data is added in Table 5. The results of hepatitis B/C were listed in lines 226-227, and 276.
Reviewer 2 Report
This observational study found that statins are associated with decreased risk of liver cancer in patients with heart failure. Please see attached pdf for comments.

Author Response
- Advice in line 65.
Response: We have revised it as suggested. - Advice in line 305
Response: We have revised it as suggested. - Advice in lines 317-323
Response:
Original content: These findings imply that metformin may diminish the chemoprotective effects of statins on liver cancer in patients with HF.
Revised content: However, after longer period of statin usage, significant lower risk was observed among patients with metformin use > 365 cDDDs (aHR 0.27, CI [0.11–0.67], aHR 0.22, CI [0.12–0.41] for statin use 91-365 cDDDs and >365 cDDDs respectively). Metformin is one of the standard treatments for patients with diabetes. It had been report-ed before that diabetes is associated with increasing risk of cancer occurrence [41]. Therefore, patients with long-duration metformin use may need longer statin use to exhibit the potential chemo-protective effect. - Advice in lines 348-350
Response:
Original content: Hence, these features potentially make rosuvastatin different from other hydrophilic statins and exhibit a strong chemoprotective effect in reducing the risk of liver cancer.
Revised content: Hence, these features potentially make rosuvastatin different from other hydrophilic statins and exhibit a strong chemoprotective effect in reducing the risk of liver cancer. However, as a high-potency statin [55], rosuvastatin may be more commonly indicated for patients with higher cholesterol levels and more severe heart failure, introducing bias by indication. Future studies are warranted to validate the findings of the present study. - Advice in line 353
Response: We have revised it as suggested. - Advice in lines 355, 359-361
Response:
Original content: First, this study was conducted using data from a health insurance claims database that lacks information on certain liver cancer risk factors, such as alcohol consumption, smoking habits, aflatoxin exposure, and body mass index. Hence, we were unable to control for these potential confounding factors. However, we used PS matching to match patients by age, sex, CCI score, diabetes, hypertension, dyslipidemia, urbanization level, and monthly income. Urbanization level and monthly income were alternative factors for lifestyle and environmental factors [25,54].
Revised content: First, this study was conducted using data from a health insurance claims database that lacks information on certain liver cancer risk factors, such as alcohol consumption, smoking habits, aflatoxin exposure, body mass index, and atherosclerosis [56,57]. Hence, we were unable to control for these potential confounding factors. However, we used PS matching to match patients by age, sex, CCI score, diabetes, hypertension, dyslipidemia, urbanization level, and monthly income. Urbanization level and monthly income were alternative factors for lifestyle and environmental factors [25,58]. Besides, CCI scores include myocardial infarction, peripheral vascular disease, and cerebrovascular disease, which have a high relationship to atherosclerosis.
- Advice in lines 377-378
Response: We have revised it as suggested.
Round 2
Reviewer 1 Report
The paper has been much improved.
Minor English editing is required.